# A Basic Study for Predicting Dysphagia in Panoramic X-ray Images Using Artificial Intelligence (AI)—Part 1: Determining Evaluation Factors and Cutoff Levels

**DOI:** 10.3390/ijerph19084529

**Published:** 2022-04-09

**Authors:** Yukiko Matsuda, Emi Ito, Migiwa Kuroda, Kazuyuki Araki

**Affiliations:** Division of Radiology, Department of Oral Diagnostic Sciences, Showa University School of Dentistry, 2-1-1 Kitasenzoku, Ohta-ku, Tokyo 145-8515, Japan; emihashimo@dent.showa-u.ac.jp (E.I.); m-kuroda@dent.showa-u.ac.jp (M.K.); araki@dent.showa-u.ac.jp (K.A.)

**Keywords:** dysphagia, panoramic radiograph, vertical hyoid bone position

## Abstract

Background: Dysphagia relates to quality of life; this disorder is related to the difficulties of dental treatment. Purpose: To detect radiographic signs of dysphagia by using panoramic radiograph with an AI system. Methods: Seventy-seven patients who underwent a panoramic radiograph and a videofluorographic swallowing study were analyzed. Age, gender, the number of remaining teeth, the distance between the tongue and the palate, the vertical and horizontal hyoid bone position, and the width of the tongue were analyzed. Logistic regression analysis was used. For the statistically significant factors, the cutoff level was determined. The cutoff level was determined by using analysis of the receiver operations characteristic (ROC) curve and the Youden Index. Results: A significant relationship with presence of dysphagia was only observed for the vertical hyoid bone position. The area under the curve (AUC) was 0.72. The cutoff level decided for the hyoid bone was observed to be lower than the mandibular border line. Conclusions: In cases where the hyoid bone is lower than the mandibular border line on a panoramic radiograph, it suggests the risk of dysphagia would be high. We will create an AI model for the detection of the risk of dysphagia by using the assessment of vertical hyoid bone position.

## 1. Introduction

The use of artificial intelligent (AI) systems is frequently attempted for the diagnosis of radiographic images. Several AI studies have been published in the maxillofacial radiographic field.

Some AI research related to the analysis of panoramic radiographs has been published previously. Tuzoff et al. [1] defined the tooth in panoramic radiographs. Ohashi et al. [2] created a diagnostic system for the large anatomical structure of the sinusitis, and Nurtanio et al. [3] classified cysts and tumor lesions. In addition to the applications to maxillofacial disease, Nakamoto et al. [4] created a screening system for osteoporosis by using panoramic radiographs. Sawagashira et al. [5] created an automatic detection system for carotid artery classifications.

Swallowing difficulties may cause the penetration and aspiration of food in the respiratory tract. Swallowing disorders are related to a high risk of aspiration pneumonia. Oral frailty can bring about poor nutrition intake and low quality of life.

Swallowing difficulties also affect dental treatments. They make it difficult to keep water in the mouth, and this is related to dysphagia aspiration, penetration, and storage in the pyriform sinus. Hou et al. [6] reported on the accidental aspiration and ingestion of foreign objects during dental treatment. The ability to predict the risk of dysphagia may improve the safety of dental treatment. If dysphagia can be predicted at an early stage, countermeasures can be taken at an early stage and frailty might be prevented.

Panoramic radiographs are commonly used in dental clinics. The advantage of radiographic images is their ability to provide an overview of the maxillofacial area and not only the teeth. Kuroda et al. [7] suggested that there is a relationship between the hyoid bone position on lateral cephalometric radiographs and panoramic radiographs. Additionally, Ito et al. [8] suggested that the hyoid bone position on panoramic radiographs is related to dysphagia.

The purpose of this study was to decide which of the radiographic signs observed on panoramic radiographs are related to dysphagia. Our results will be applied for the creation of an AI program.

## 2. Materials and Methods

### 2.1. Study Design

This study is a retrospective study that analyzes patient data previously tested and collected. Data correction was planned after the panoramic radiographic exam and videofluorographic (VF) exam.

### 2.2. Participants

Patient data were collected between July 2013 and March 2021. A total of 1155 VF-examined patients were selected. In these patients, surgical cases due to tumors and jaw deformity were excluded. Following this, 476 patients remained.

Patients who had undergone panoramic radiography were assessed. The exclusion criteria of panoramic radiographs, which were related to the positioning error, were in accordance with Izzetti et al. [8]. These panoramic radiographs were analyzed for symmetry of image, inclination of occlusal plane, and localization of mandibular condyles. Patients who were unable to maintain correct positioning during panoramic radiography and cases where the hyoid bone moved during panoramic radiography were also excluded. 

Finally, 77 patients who underwent both a VF exam and a panoramic radiographic exam were selected. 

### 2.3. Panoramic Radiography

Panoramic radiographs were taken by a Hyper-XF radiography machine (Asahi Roentgen Ind. Co., Ltd., Kyoto, Japan). The exposure parameters were set at 78 to 82 kV, 10 mA, and 12 s. Head positioning in panoramic radiographs used a standardized protocol. During the panoramic radiograph, the patient bite down on a cotton roll to prevent infection. The patients were also instructed to relax their tongues.

### 2.4. Image Assessment and Measurements

These panoramic radiographs were assessed using a Windows computer, HP Compaq 6300 Pro (Hewlett-Packard, Palo Alto, CA, USA), with a 21.1 inch, 2M pixel medical color LED monitor RX240 (EIZO, Tokyo, Japan).

Two oral and maxillofacial radiologists, who did not know the results of the VF exam, assessed the images. In cases where different diagnoses were obtained, the images were re-examined, and consensus was reached between both experts.

Evaluation of the vertical hyoid bone position, the horizontal hyoid bone position, and the measurement of the distance from the tongue to the palate were performed according to the method of Ito et al. [9]. The measurement and evaluation methods are shown below.

#### 2.4.1. Vertical Hyoid Bone Position

Figure 1 illustrates the evaluation of the vertical position of the hyoid bone and shows sample images. This method of measurement was as specified in Ito et al. [9]. Figure 2 shows sample images.

Two landmarks were defined, as follows:The bilateral mandible line: A simulated line connecting the right and left sides of the angles of the mandible.The mandibular border line: The line that moved the bilateral mandibular line parallel to the lowest point of the lower border of the mandible.

An evaluation was conducted of the extent to which the hyoid body and greater horn appeared in the upper area from the mandibular border line. The following six groups were categorized:

Type 0: The hyoid bone could not be observed in the upper area from the mandibular border line;

Type 1: Only the greater horn was observed in the upper area from the mandibular border line;

Type 2: A less than half of the hyoid body was observed in the upper area from the mandibular border line;

Type 3: More than half and less than whole of the hyoid body was observed in the upper area from the mandibular border line;

Type 4: All of the hyoid body was observed in the upper area from the mandibular border line;

Type 5: The hyoid body overlapped with the mandible bone.

On the right and left sides, if the vertical position of the hyoid bone was different, the lower position side was recorded.

#### 2.4.2. Horizontal Hyoid Bone Position

Figure 3 illustrates how to evaluate the position of the horizontal hyoid bone and Figure 4 shows sample images. This method of measurement was taken from Ito et al. [9]. The horizontal position of the hyoid bone was graded based on the anterior point of the hyoid body. Cases in which the hyoid bone was invisible were excluded from this assessment.

Six landmarks were defined, as follows:The mesial premolar line is the point between the mesial interproximal surface of the first premolar and the vertical border of the mandible until the end of the image. When the first premolar was missing, the distal surface of the canine was used.The distal premolar line is the point between the distal interproximal surface of the second premolar and the vertical border of the mandible until the end of the image. When the second premolar was missing, the mesial surface of the first molar was used.The premolar area is the area between the mesial premolar line and the distal premolar line.The distal molar line is the perpendicular line from the distal interproximal surface of the second molar or the mesial interproximal surface of the third molar to the edge of the image through the lower border of the mandible. When the second molar was missing, the mesial surface of the third molar was used.The molar area is the area between the distal premolar line and the distal molar line.The posterior area is the area posterior to the distal molar line.

In cases where the relevant tooth or its neighboring tooth was missing, these landmark lines were referred to the maxillary tooth.

Grade 1 is the most anterior point of the hyoid body that is observed in the premolar zone.

Grade 2 is the most anterior point of the hyoid body that is observed in the molar zone.

Grade 3 is the most anterior point of the hyoid body that is observed in the posterior zone.

The detailed technical description of these measures and their assessments (acquisition) has been previously reported and summarized.

#### 2.4.3. Measurement from the Tongue to the Palate on the Midline (mm)

Figure 5 shows the sample image of measurement from the tongue to the palate. This method of measurement was taken from Ito et al. [9]. The distance of the tongue to the palate on the midline was measured. The midline is the line that connects the anterior nasal spines to the interproximal point of the maxillary incisors. On the midline, the distance from the surface of the tongue to the palate was measured.

#### 2.4.4. Width of Tongue

Figure 6 illustrates how to evaluate the outer position of tongue and Figure 7 shows sample images. The width of tongue was assessed according to the location where the outer border of tongue overlapped the anatomical structure. If the right side and left side of the outer border of the tongue were different, the inner position side was used.

Three landmarks were defined, as follows:

The distal border of the mandible line is a simulated line, which is defined between the posterior point of the mandibular condyle and posterior point of the angle of the mandible.The mesial border of the mandible line is defined as the line that is the distal border line moved in parallel to the coronoid process.The midline of the border of the mandible line is defined as the midline of the distal border of the mandible line and the mesial border of the mandible line.

The outer border of tongue is defined according to the following positions:

Position 1: Observed between the mesial border of the mandible line and the midline border of the mandible line;

Position 2: Observed between the midline of border of the mandible line and the distal border of the mandible line;

Position 3: Observed on the distal border of the mandible line;

Position 4: Observed outside of the distal border of the mandible line.

### 2.5. Classified Dysphagia (+) and Dysphagia (−) Groups

The protocol for the VF of each patient involved presentation of swallows with 1 to 3 mL standard meals with barium without any instruction for swallowing. All three or four dentists involved in the examination inspected laryngeal invasion, aspiration, and accumulation of epiglottis valley/pyriform sinus. At least two of the three or four observers were qualified as Board Certified Fellows of the Japanese Society of Dysphagia Rehabilitation. The examiners in the VF room can miss the findings on the monitor because they are concentrating on the patient’s condition. In our hospital, one or two dentists are in the VF room, and the others are outside the VF room, checking the monitor, to obtain consensus. Figure 8 shows the sample VF images of aspiration (left image) and penetration (right image).

Observers recorded these findings on the VF exam chart. These VF exam charts were used for the classification of a dysphagia (+) case and (−) case. In this study, penetration and aspiration cases were treated as a dysphagia (+) case.

### 2.6. Statistical Analysis

#### 2.6.1. Logistic Regression Analysis

SPSS Statistics Ver. 27.0 (International Business Machines Corporation, Chicago, IL, USA) was used.

The logistic regression test was used to analyze the independent influencing prognostic factors of the risk of dysphagia. A two-sided *p*-value of <0.05 was considered significant. Differences were analyzed between the dysphagia (+) and dysphagia (−) groups in age, gender, number of remaining teeth, vertical and horizontal hyoid bone position, the distance of the tongue to the plate, and the width of the tongue.

A correlation matrix was created in advance when the independent variables were input, and it was confirmed that there was no strong correlation between the independent variables with r > 0.80.

#### 2.6.2. Decision of Cutoff Level

The effective factor, which was defined by logistic regression analysis, a receiver operating characteristic curve (ROC curve), and the decision on the cutoff level, was achieved using JMP Pro Ver. 16.0 (SAS Institute Inc., Cary, NC, USA). Cutoff level was determined using the Youden Index. It consisted of the maximum number of Sensitivity-(1-Specificity).

## 3. Results

Based on the results of the VF findings, 47 patients were diagnosed with dysphagia. These patients were classified into the dysphagia (+) group. The remaining 30 patients were not diagnosed with dysphagia. These patients were classified into the dysphagia (−) group.

Table 1 shows the clinical characteristics of the patient groups divided by the presence of dysphagia using the VF exam. Table 2, Table 3 and Table 4 show the prevalence of the vertical positions of the hyoid bone, the horizontal hyoid bone positions, and the tongue widths. In Table 3, the invisible hyoid bone cases were excluded, resulting in 23 cases of dysphagia (+) and 3 cases of dysphagia (−).

The correlation matrix between independent variables was assessed. It was confirmed that there is no strong correlation between the independent variables with r > 0.80.

Table 5 shows the predictors of dysphagia (+) in a stepwise logistic regression analysis. An independent influencing factor of dysphagia was the vertical hyoid bone position (*p* = 0.005, beta was 0.448 and Exp(B) was 1.56). The odds ratio was 1.506, which suggested that if the vertical hyoid bone position grade was changed to 1–0, the risk of dysphagia would become 1.506.

ROC analysis was performed for the vertical hyoid bone position. Figure 9 shows the ROC curve of the vertical hyoid bone position. The number of the Area Under Curve (AUC) is 0.715. This suggests that it has moderate accuracy. Using the ROC results, the cutoff level was analyzed. Table 6 shows the results of the Youden Index. The maximum number of the Sensitivity-(1-Specificity) was Grade 0. Grade 0 suggests the most accurate grade for the risk of dysphagia.

## 4. Discussion

### 4.1. Hyoid Bone Position

The larynx and hyoid bone complex is one of the most important factors for swallowing without aspiration. Logemann et al. analyzed the swallowing of young and old females [10] and males [11]. They concluded that reduced hyoid elevation is a common cause of aspiration. Perlman et al. [12] suggested that reduced hyoid elevation was associated with the risk of aspiration. In our study, the vertical hyoid bone’s lower position was related to dysphagia. When the hyoid bone is in a lower position, with reduced hyoid elevation, it takes time to move to the swallowing position, which can be one of the causes of dysphagia.

There is no paper that evaluates the position of the hyoid bone using panoramic radiographs. This may be due to the variation of the hyoid bone position. Most of the papers on the evaluation of the position of the hyoid bone are analyses of lateral cephalometric radiographs [13,14], VF images [10,11,15,16,17,18], and computed tomography (CT) [19]. Studies have performed measurements by using cephalometric images and CT, and otherwise evaluated relative positional relationships by using VF images in the assessment of hyoid bone position.

We determined the cutoff value for discriminating cases with dysphagia. In the future, we aim to screen for dysphagia and detect the risk of dysphagia at an early stage using panoramic radiographs. In this study, the patients with suspected dysphagia were analyzed. The cutoff level was determined as Grade 0, which means the hyoid bone is lower than the mandible line. This position was too low. One of the reasons for this is that it was based on those people with suspected dysphagia. If it were possible to include normal people with no dysphagia symptoms, who are confirmed not to have dysphagia, the cutoff level might be shifted to a higher level on panoramic radiographs.

No relation was found between the horizontal hyoid bone position and dysphagia. However, several research papers have been published regarding hyoid bone position analysis using VF examination. Kim et al. [15] analyzed the changing position of the hyoid bone vertically and horizontally in normal swallowing by VF exam. They concluded that there was a significant difference between younger and older people in the anterior displacement of the hyoid bone during swallowing. They also concluded that the hyoid bone did not show a vertical displacement, and they found no significant differences between males and females. Anterior displacement of the hyoid bone was decreased with age. This reduction may be related to muscle weakness. Uarguchi et al. [16] concluded that the mechanism of dysphagia was revealed by structural equation modeling, indicating that insufficient anterior movement could lead to pharyngeal residue in the pyriform sinus. Zhang et al. [16] concluded that anterior-horizontal hyoid bone displacement was the important factor in predicting penetration and aspiration risk. In addition, Paik et al. [18] concluded that horizontal moving of the hyoid bone and rotation of the epiglottis were reduced in patients with myopathy as compared to controls and patients with cerebral infarction (*p* < 0.05). Patients with dysphagia showed different patterns as compared to controls in trajectory analysis according to their etiology. They concluded that the extent and pattern of movement of the hyoid bone and the epiglottis during swallowing were different according to the etiology of dysphagia, and swallowing motion analysis could be applied to differentiate the mechanism of dysphagia. Several VF researchers suggested that the hyoid bone shifted in the posterior but not the vertical direction. The reason for this difference may be the modality of radiographic images, such as head position.

Feng et al. [19] assessed the hyoid bone position by analyzing CT images in healthy older adults and aspiration patients. They concluded that the distance from the hyoid bone to the border of the mandible is increased with aging and found a larger posterior position in aspiration patients. When taking CT images, patients lay down, and the hyoid bone could more easily shift to the posterior position. These results suggest that in the aspiration patients in comparison with the normal patients, the weakness of hyoid muscles might be related to the difficulty to maintain the normal hyoid bone position. If these studies were to add analysis of the vertical hyoid bone position with cephalometric radiograph or panoramic radiographs, the hyoid bone position of aspirated patients may be observed to be lower than in healthy elderly people.

In this study, we analyzed which radiographic signs are related to dysphagia. We decided to focus on the vertical hyoid bone position, and the cutoff level was Grade 0. By using these results, we plan to create an AI program for the detection of the risk of dysphagia.

Several tongue training effects have been reported to be useful in preventing dysphagia [20,21,22]. By creating an AI program for screening for dysphagia using panoramic radiographs, we aim to help in the early detection and prevention of dysphagia and sarcopenia and help improve the safety of dental treatment.

### 4.2. Size of Tongue

In this study, the distance from the tongue to the palate was measured, and the outer border of the tongue was assessed. If the size of tongue is small, this might suggest a reduced hyoid muscle, and thus suggest dysphagia. However, no significant factors were found. Dysphagia thus may not be related to the size of the tongue. Nakao et al. [19] reported similar results. They analyzed the associations between age, tongue muscle, tongue pressure, and predysphagia by using MRI images. They concluded that the tongue pressure did not relate to the size of tongue. Additionally, tongue fat mass and percentage increased with age; the tongue fat percentage in elderly participants was 20%, which was two times larger than that of young participants.

### 4.3. Sample Size

In the case of the chi-square test, the degree of freedom (df) is 5 for the vertical position of the tongue. If the effect size is 0.5, which is large (EF), the sample size is 52. The sample size for this study is 77. If the effect size (EF) is 0.3, the sample size is 143. From the above, the substantial difference in this study lies between medium and large.

### 4.4. Limitations of This Study

The gold standard is a VF exam. The objects of this study were suspected dysphagia patients in medical clinics and nursing homes. These patients differed from normal persons. If we could include normal patients who do not have dysphagia, the cutoff level of the vertical hyoid bone position might be shifted to a higher position. However, it is difficult to perform VF exams with normal persons. It is impossible to perform VF analysis without any symptoms of dysphagia, and this is a limitation of this study.

In this study, the position of the hyoid bone in the panoramic radiograph was evaluated. In our hospital, the patient was instructed to gently bite cotton roll and relax their tongue to avoid their incisor teeth overlapping. Izzetti et al. [8] suggested that the patients should have to keep the tongue pressed against the palate, but it is difficult to keep the distance between the upper and lower anterior tooth with the tongue pressed to the palate. Thus, the relaxed tongue position was used.

There is no report on the evaluation of the hyoid bone position and size of tongue on the panoramic image. That is one of the limitations of this study. It may be possible to take panoramic radiographs with the hyoid bone intentionally raised or lowered in healthy people. However, keeping the raised position for more than 12 s of exposure time is difficult for people with dysphagia. That is the reason why the hyoid bone movement was excluded this study. We need a simple method for how to evaluate this for the creation of an AI program. Regarding the position of the hyoid bone, we referred to the measurement of the hyoid bone in the lateral cephalometric radiograph and considered whether there is a method that could be easily evaluated. In this study, we considered evaluating the position of the hyoid bone based on the lower border of the mandible. In addition, there is no standard assessment method for the size of the tongue. The size of the tongue on the panoramic image was evaluated according to the distance from the tongue to the palate, and the lateral surface of the tongue overlapped with the ramus of the mandible. The tongue is a soft tissue. Swallowing is done in the sitting position, not in the supine position. Panoramic radiographs are taken in a sitting or standing position, but CT and MRI examinations are generally performed in a lying position, so it is difficult to compare these 3D methods. Only sitting MRIs and CTs may solve this problem.

The size of the tongue on the panoramic image was assessed by two points: the vertical one, depending on the distance from the surface of the tongue to the palate on the midline, and the degree to which the lateral surface of the tongue overlaps with the ramus of the mandible. If there was a significant difference, we planned to use this evaluation as an evaluation point for AI. However, there were no significant.

We considered the hyoid bone position as stable, because the hyoid bone did not move during the 12 s of taking the panoramic radiograph. Since the positioning of panoramic radiographs is not strictly standardized, as is the case for lateral cephalometric radiographs, there is some variation in our results. It was impossible to perform lateral cephalometric radiographs for all patients with suspected dysphagia, and this is also a limitation of this study. The panoramic radiographic exam is frequently used in dental clinics rather than cephalometric radiographs or VF exams.

## 5. Conclusions

Our results indicate that the factor that is related to dysphagia is the vertical hyoid position in panoramic radiographs. If the hyoid bone was lower than the mandibular border line, the risk of dysphagia was 1.5 times higher. In the future, we will build an AI model based on this factor.

## Figures and Tables

**Figure 1 ijerph-19-04529-f001:**
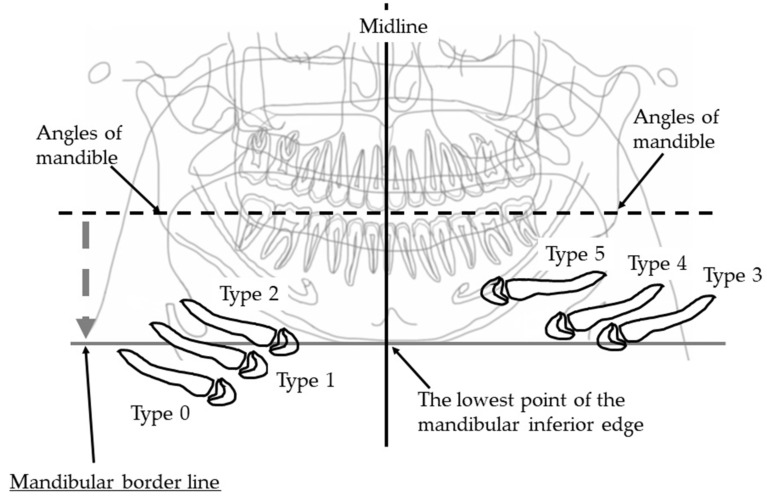
Vertical hyoid bone position. Mandibular border line was defined as a line that moves virtual line which is connecting the both side of mandibular angles, in parallel along the center line and attach to the lowest point of the mandibular inferior edge. An evaluation was conducted of the extent to which the hyoid body and greater horn appeared in the upper area from the mandibular border line.

**Figure 2 ijerph-19-04529-f002:**
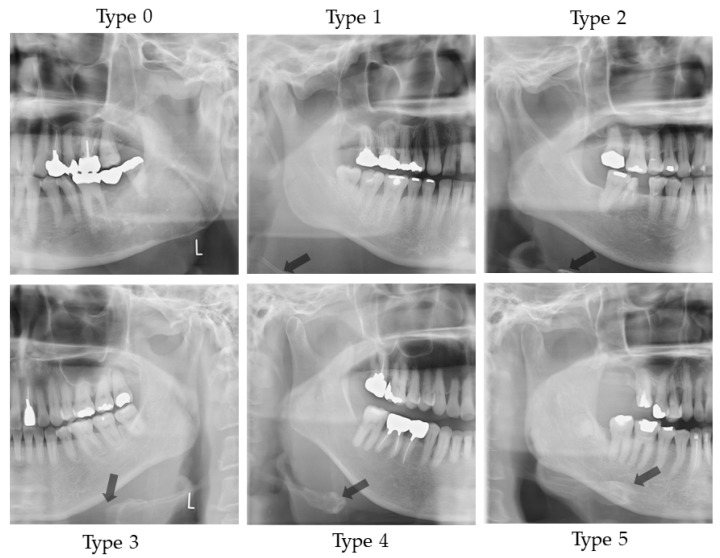
Sample images of vertical hyoid bone position. Arrow shows the hyoid bone. On Type 0, hyoid bone is invisible.

**Figure 3 ijerph-19-04529-f003:**
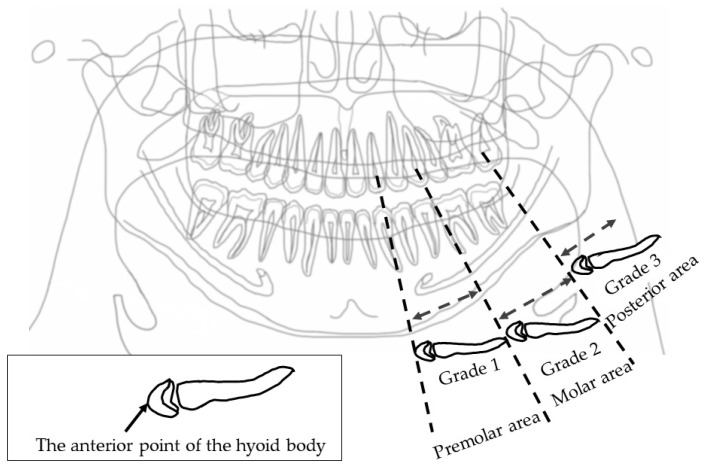
Horizontal hyoid bone position. The horizontal hyoid bone position was graded by the position of the anterior point of the hyoid body.

**Figure 4 ijerph-19-04529-f004:**
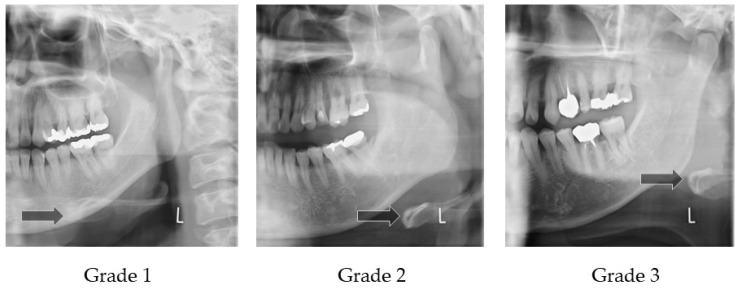
Sample images of horizontal hyoid bone position. Arrow shows the anterior point of the hyoid body.

**Figure 5 ijerph-19-04529-f005:**
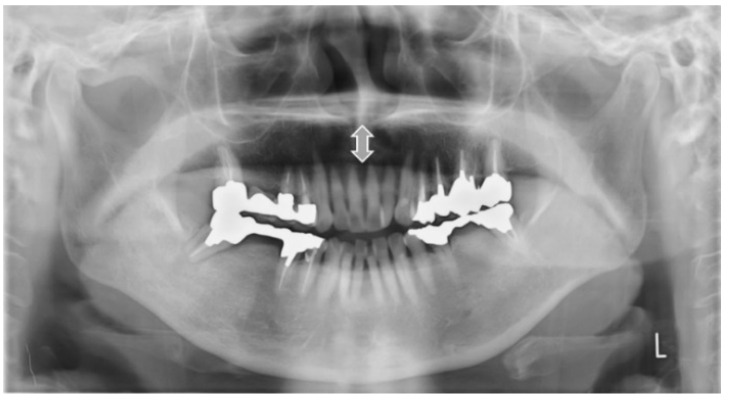
Measurement from the surface of the tongue to the palate on the midline (mm). Double sided arrow shows the measurement between the palate to tongue.

**Figure 6 ijerph-19-04529-f006:**
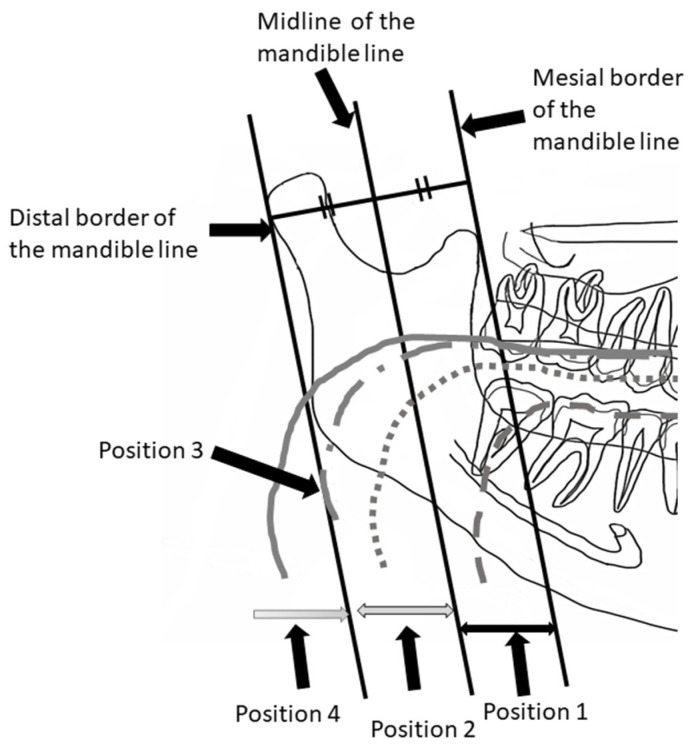
Width of the tongue. The location where the outer border of tongue overlapped the anatomical structure was assessed.

**Figure 7 ijerph-19-04529-f007:**
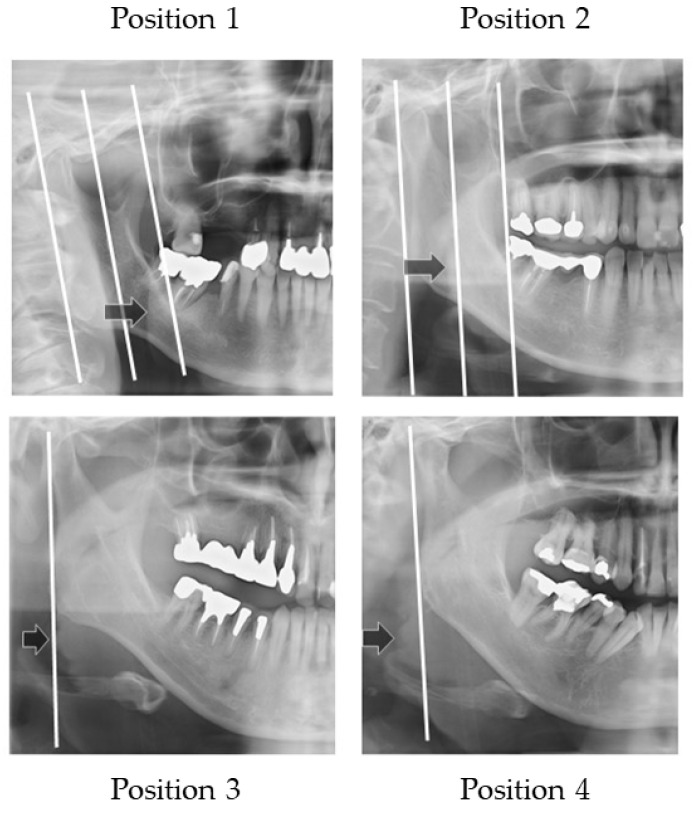
Sample images of width of tongue. Arrows show the outer border of the tongue.

**Figure 8 ijerph-19-04529-f008:**
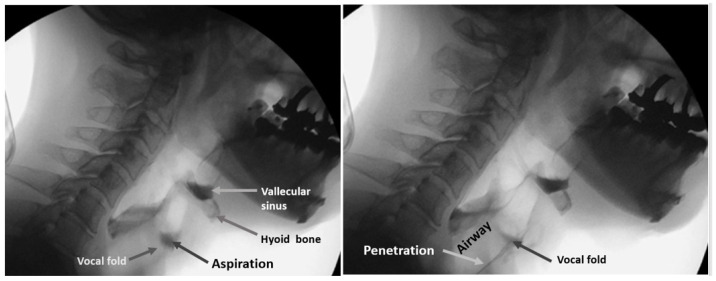
Sample VF images. Left image shows the aspiration and right image shows penetration.

**Figure 9 ijerph-19-04529-f009:**
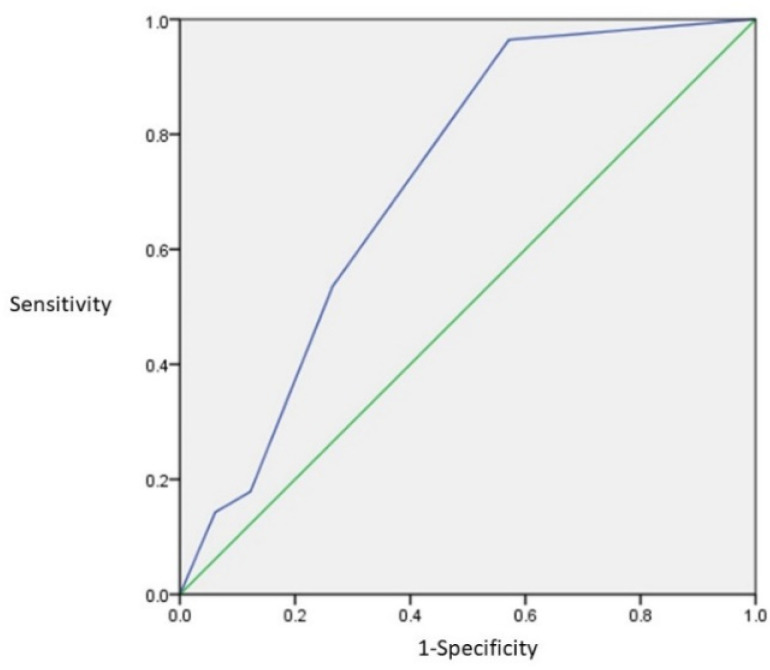
The ROC curve of the vertical hyoid bone position. The AUC was 0.715.

**Table 1 ijerph-19-04529-t001:** Clinical characteristics of patient groups divided by presence of dysphagia by VF exam.

	Dysphagia (+), n = 47	Dysphagia (−), n = 30
Age (years)	77.0 (72.4–81.6)	68.5 (62.6–74.4)
Gender male (%)	33 (70.0)	18 (60.0)
Number of remaining teeth	22.0 (19.3–24.7)	24.0 (20.9–27.1)
Distance from hyoid to palate	12.5 (9.6–15.4)	11.4 (7.8–5.1)

Median (95% confidence interval for median).

**Table 2 ijerph-19-04529-t002:** Prevalence of the vertical hyoid bone position.

Vertical Hyoid Bone Position	Dysphagia (+)	Dysphagia (−)
Type 0	20	2
Type 1	10	8
Type 2	4	5
Type 3	7	10
Type 4	3	1
Type 5	3	4

**Table 3 ijerph-19-04529-t003:** Prevalence of the horizontal hyoid bone position.

Horizontal Hyoid Bone Position	Dysphagia (+)	Dysphagia (−)
Grade 1	5	9
Grade 2	15	15
Grade 3	4	3

**Table 4 ijerph-19-04529-t004:** Prevalence of the horizontal position of hyoid bone.

Outer Border of Tongue	Dysphagia (+)	Dysphagia (−)
Position 1	4	3
Position 2	18	8
Position 3	13	12
Position 4	12	7

**Table 5 ijerph-19-04529-t005:** Predictors of dysphagia (+) in a logistic regression analysis.

	β–Coefficient	Dysphagia (+) 95% Confidence Intervals	*p*-Value
Age	0.002	1.002 (0.779–1.289)	0.989
Gender	2.4	11.095 (0.653–188.607)	0.096
Number of remaining teeth	−0.16	0.984 (0.899–1.076)	0.720
Vertical hyoid bone position	0.694	2.002 (1.024–3.912)	0.042 *
Horizontal hyoid bone position	−0.123	0.885 (0.385–2.031)	0.772
Distance hyoid to palate	0.079	1.082 (0.999–1.171)	0.053
Width of tongue	−0.143	0.867 (0.581–1.293)	0.484

* Statistically significant.

**Table 6 ijerph-19-04529-t006:** Results of Youden Index (Sensitivity − (1 − Specificity)) and probability.

Grade	Prob.	1 − Spec.	Spec.	Sensi. − (1 − Spec.)	TP	TN	FP	FN
		0.0000	0.0000	0.0000	0	28	0	49
0	0.8052 *	0.0357	0.4286	0.3929 *	21	27	1	28
1	0.7253	0.3214	0.6327	0.3112	31	19	9	18
2	0.6277	0.4643	0.7347	0.2704	36	15	13	13
3	0.5185	0.8214	0.8776	0.0561	43	5	23	6
4	0.4075	0.8571	0.9388	0.0816	46	4	24	3
5	0.3052	1.0000	1.0000	0.0000	49	0	28	0

* The maximum number of the Sensitivity − (1 − Specificity) and probability. Grade 0 suggests the most accurate grade for the risk of dysphagia. Prob.: probability; Spec.: specificity; Sensi.: sensitivity; TP: true positive; TN: true negative; FP: false positive; FP: false negative.

## Data Availability

Not applicable.

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
