# Peer review of "A Basic Study for Predicting Dysphagia in Panoramic X-ray Images Using Artificial Intelligence (AI)—Part 1: Determining Evaluation Factors and Cutoff Levels"

_ijerph, 2022, doi:10.3390/ijerph19084529_

Round 1
Reviewer 1 Report
It is an interesting report; however, some unreported methodological aspects inhibit a detailed assessment of the reproducibility and applicability of the results. My comments were as follows: 1. The description of the study design is missing. Since metrics from a diagnostic study are reported, I suggest incorporating all possible items from the STARD guideline (see Bossuytet al. STARD 2015: An Updated List of Essential Items for Reporting Diagnostic Accuracy Studies. Radiology, 277 (3), 826-832. ); 2. Explain the assumptions for using the sample size; 3. The ROC curves should be provided; 4. I am unclear whether the test dataset was truly independent: was it from the same centre?
Author Response
Dear Reviewer 1,
Thank you for your peer review. There has been no report on how to evaluate the hyoid bone and tongue on panoramic images, and I was struggling with how to evaluate it. Basically, I referred to the evaluation method of the hyoid bone in the lateral cephalometric measurement and considered whether there is a method that can be easily evaluated. As a result, we considered evaluating the position of the hyoid bone based on the lower border of the mandible. Also, there is no standard assessment method for the size of the tongue. The size of the tongue on the panoramic image was set to two points: the vertical one depending on the distance from the back of the tongue to the palate, and the horizontal size was assessed by degree to how much the side surface of the tongue overlaps with the ramus of the mandible. We need a simple method on how to evaluate them for the creation of AI program. If research on tongue size using panoramic radiograph comes out, I would like to evaluate it by that method. However, the tongue is a soft tissue. Swallowing is done in the sitting position, not in the supine position. Panoramic radiographs are taken in a sitting or standing position, but CT and MRI examinations are generally performed in a lying position, so it is highly likely that the shape and position of the tongue will differ. Also, this may be the limitation of this study and no one did not analyze.
This study is a retrospective study that analyzes the patient data that has already been tested and collected.
When I reviewed my article, there were many points where the explanation was insufficient when conducting new research, so I added sentences such as the report shown for reference and the explanation of the research design based on STARD 2015.
- Study design: It is a retrospective study. We first checked the 1155 patients who was taken VF exam. Exclusion criteria, the person who was taken the panoramic radiograph, 476 patients were remaining. Check the quality of panoramic radiograph, finally 77patients panoramic radiographs were assessed.
- Sample size: In the case of the χ-square test, the degree of freedom (df) is 5 for the vertical position of the tongue. If the effect size is 0.5, which is large effect (EF), the sample size is 52. The sample size for this study is 77. If the effect size (EF) is 0.3, the sample size is 143, and the substantial difference in this study is considered to be between medium to large. 3. "ROC Graph" has been added.
- About "I don't know if the test datasets are really independent." : The judgment of the test data(panoramic radiograph) and the judgment of the gold standard set(result of VF exam) were made separately by the radiology department and the oral rehabilitation department, and the information of both was not exchanged in advance.
I would appreciate it if you could confirm the correction contents.
Best regards
Yukiko Matsuda

Reviewer 2 Report
The manuscript evaluates the role of panoramic radiography in predicting dysphagia. The authors selected the parameters of hyoid bone position and tongue volume. There are however some issues which need clarification:
- Were the panoramic radiographs acquired using a standardized protocol? Were variations in patient positioning taken into account?
- It can be observed form the images provided that the occlusal plane largely varies in curvature, accounting for a variability in patient positioning (see Appl. Sci. 2021, 11(17), 7858; https://doi.org/10.3390/app11177858). This should be listed as a major limitation to the study, and could potentially affect the validity of the results. The authors should clarify this fact.
- It would be useful to add videofluorographic swallowing study images of the patients.
Author Response
Dear Reviewer 2,
Thank you for your advice. Panoramic radiograph was compared with the content of the review by Izzetti et al et al., which was shown for reference. The positioning for the device was the same, but the patient's occlusion and tongue position instructions at the time of imaging were different. When taking panoramic radiograph, the patient was bite with a cotton roll, for prevent infection. Also, the patient was instructed to relax his tongue. This is a big difference from the content of the review you instructed. At our hospital, from the viewpoint of infection prevention, we gently bite the cotton roll.
If strongly bite, the front teeth of the upper and lower jaws will overlapped. This method is different from conventional taking panoramic radiograph, but it was more important thing to prevent overlapping anterior teeth. In that respect it was not a standardized method.
It may be possible to take panoramic radiographs with the hyoid bone intentionally raised or lowered in healthy people. However, keeping the raised position for more than 10 seconds of shooting time is difficult for people with dysphagia. That is a reason for the hyoid bone movement was excluded this study.
A reference VF test images will be helped for the results.
Best regards,
Yukiko Matsuda

Round 2
Reviewer 1 Report
Thank you for providing the satisfactory revision.
Reviewer 2 Report
The issues raised in the previous revision were all solved. There are no further comments.